# Importance Resampling for Off-policy Policy Evaluation

## Abstract

Importance sampling is a common approach to off-policy learning in reinforcement learning. While it is consistent and unbiased, it can result in high variance updates to the parameters for the value function. Weighted importance sampling (WIS) has been explored to reduce variance for off-policy policy evaluation, but only for linear value function approximation. In this work, we explore a resampling strategy to reduce variance, rather than a reweighting strategy. We propose Importance Resampling (IR) for off-policy learning, that resamples experience from the replay buffer and applies a standard on-policy update. The approach avoids using importance sampling ratios directly in the update, instead correcting the distribution over transitions before the update. We characterize the bias and consistency of the our estimator, particularly compared to WIS. We then demonstrate in several toy domains that IR has improved sample efficiency and parameter sensitivity, as compared to several baseline WIS estimators and to IS. We conclude with a demonstration showing IR improves over IS for learning a value function from images in a racing car simulator.

## 1 Introduction

An important component of many learning systems is learning value functions for many policies. Some examples of such systems are the Horde architecture composed of General Value Functions (GVFs) (Sutton et al., 2011; Modayil et al., 2014), systems that use options (Sutton et al., 1999; Schaul et al., 2015a), predictive representation approaches (Sutton et al., 2005; Schaul and Ring, 2013; Silver et al., 2017) and systems with auxiliary tasks (Jaderberg et al., 2017). For each target policy, the value function returns the expected return from a state, and can provide useful information about (long-term) outcomes under different behaviors. Off-policy learning is critical for learning many value functions with different policies at scale, because it enables data to be generated from one behavior policy to update the values for each target policy in parallel.

The typical strategy for off-policy learning is to use importance sampling (IS). For a given state $s$, with action $a$ selected according to behaviour $\mu$, the importance sampling ratio is the ratio between the probability of the action under the target policy $\pi$ and the behaviour: $\frac{\pi(a|s)}{\mu(a|s)}$. The update is multiplied by this ratio, adjusting the action probabilities so that the expectation of the update is as if the actions were sampled according to the target policy $\pi$. Though the IS estimator is unbiased and consistent (Kahn and Marshall, 1953; Rubinstein and Kroese, 2016), it can suffer from high or even infinite variance due to large magnitude IS ratios, in theory (Andradottir et al., 1995) and in practice (Precup et al., 2001; Mahmood et al., 2014; 2017).

There have been some attempts to modify policy evaluation algorithms to mitigate this variance.[1] Weighted IS (WIS) algorithms have been introduced (Precup et al., 2001; Mahmood et al., 2014; Mahmood and Sutton, 2015), which normalize each update by the sample average of the ratios. These algorithms did improve learning over standard IS strategies, but are not straightforward to extend to nonlinear function approximation. In the offline setting, a reweighting scheme, called importance sampling with unequal support (Thomas and Brunskill, 2017), was introduced to account

---

[1]There is substantial literature on variance reduction for another area called off-policy policy evaluation, but which estimates only a single number or value for a policy (e.g., see (Thomas and Brunskill, 2016)). The resulting algorithms differ substantially, and are not easily applicable for learning the value function.

for samples where the ratio is zero, in some cases significantly reducing variance. Another strategy has been to use rescaling or truncation of IS ratios, such as in V-trace (Espeholt et al., 2018). Several other methods have introduced bias similarly for algorithms with eligibility traces by truncating or scaling IS ratios to maintain stability of the eligibility trace vector, including Tree-Backup (Precup et al., 2000), Retrace (Munos et al., 2016) and ABQ (Mahmood et al., 2017). Truncation of IS-ratios in V-trace can incur significant bias, and this additional truncation parameter does need to be tuned.

An alternative to reweighting updates is to instead correct the distribution before updating the estimator using weighted bootstrap sampling: resampling a new set of data from the previously generated samples (Smith et al., 1992; Arulampalam et al., 2002). Consider a setting where a buffer of data is stored, generated by a behavior policy. Samples for policy $\pi$ can be obtained by resampling from this buffer, proportionally to $\frac{\pi(a|s)}{\mu(a|s)}$ for state-action pairs $(s, a)$ in the buffer. In the sampling literature, this strategy has been proposed under the name Sampling Importance Resampling (SIR) (Rubin, 1988; Smith et al., 1992; Gordon et al., 1993), and has been particularly successful for Sequential Monte Carlo sampling (Gordon et al., 1993; Skare et al., 2003). Such resampling strategies have also been popular in classification, with over-sampling or under-sampling typically being preferred to weighted (cost-sensitive) updates (Lopez et al., 2013).

Such a resampling strategy, however, has yet to be proposed for policy evaluation, though there are several potential benefits. By correcting the distribution before updating, standard on-policy updates can be used, without needing to modify or re-derive with IS ratios. This simplifies application of different optimizers, such as those with momentum terms. Another benefit is that the magnitude of the updates will vary less—because updates are not multiplied by very small or very large importance sampling ratios—potentially reducing variance of stochastic updates and simplifying stepsize selection. Resampling should have larger benefits for learning approaches, as compared to averaging or numerical integration problems, because updates accumulate in the weight vector and change the optimization trajectory of the weights. For example, very large importance sampling ratios could destabilize the weights. Such a problem does not occur for resampling, as instead the same transition will simply be resampled multiple times, spreading out the large magnitude update across multiple updates. On the other extreme, with small ratios, IS will waste updates on transitions with very small IS ratios. Resampling, therefore, should have better sample efficiency. Two important questions, therefore, are if these hypothesized advantages manifest in practice in off-policy learning, and how SIR can be extended for use in policy evaluation.

In this work, we investigate the use of SIR for online off-policy policy evaluation. We first introduce Importance Resampling (IR), which uses an SIR strategy to sample transitions from a buffer of (recent) transitions. These sampled transitions are then used for on-policy updates. We show that IR, with a sliding-window buffer, is a consistent estimator of the one-step on-policy updates, with the same bias as WIS. We then empirically investigate IR on three toy domains and a racing car simulation learning from images. We find that IR is more sample efficient—learning more quickly in terms of number of updates—and has reduced sensitivity in terms of learning rate parameters, for both fixed parameters and within the RMSProp optimizer.

## 2 BACKGROUND

We consider the problem of learning General Value Functions (GVFs) (Sutton et al., 2011). The agent interacts in an environment defined by a set of states $\mathcal{S}$, a set of actions $\mathcal{A}$ and Markov transition dynamics, with probability $\mathrm{P}(s'|s, a)$ of transitions to state $s'$ when taking action $a$ in state $s$. A GVF is defined for policy $\pi : \mathcal{S} \times \mathcal{A} \rightarrow [0, 1]$, cumulant $c : \mathcal{S} \times \mathcal{A} \times \mathcal{S} \rightarrow \mathbb{R}$ and continuation function $\gamma : \mathcal{S} \times \mathcal{A} \times \mathcal{S} \rightarrow [0, 1]$, with $c_{t+1} \stackrel{\text{def}}{=} c(S_t, A_t, S_{t+1})$ and $\gamma_{t+1} \stackrel{\text{def}}{=} \gamma(S_t, A_t, S_{t+1})$, with values

$$V(s) \stackrel{\text{def}}{=} \mathbb{E}_\pi \left[ c_{t+1} + \gamma_{t+1} c_{t+2} + \gamma_{t+1} \gamma_{t+2} c_{t+3} + \dots | S_t = s \right] = \mathbb{E}_\pi \Big[ \sum_{i=1}^{\infty} \Big( \prod_{j=0}^{i-1} \gamma_{t+j} \Big) c_{t+i} | S_t = s \Big].$$

The operator $\mathbb{E}_\pi$ indicates the actions are selected according to policy $\pi$ for the expectation. GVFs encompass standard definitions of value functions, where the cumulant is a reward and the continuation function is a constant. Otherwise, they specify a broader set of value functions, that enable predictions about discounted sums of others signals into the future, when following a target policy $\pi$. These values are typically estimated using parametric function approximation, with parameters $\theta \in \mathbb{R}^d$ defining approximate values $V_\theta(s)$.

In off-policy learning, transitions are sampled according to behaviour policy, rather than the target policy. To get an unbiased sample of an update to the parameters, the action probabilities need to be adjusted. Consider on-policy temporal difference (TD) learning, with update $\alpha_t \delta_t \nabla_\theta V_\theta(s)$ for a given $S_t = s$, for stepsize $\alpha_t \in \mathbb{R}^+$ and TD-error $\delta_t \overset{\text{def}}{=} C_{t+1} + \gamma_{t+1} V_\theta(S_{t+1}) - V_\theta(s)$. If actions are instead sampled according to a behaviour policy $\mu : \mathcal{S} \times \mathcal{A} \to [0, 1]$, then we can use importance sampling (IS) to modify the update, giving the off-policy TD update $\alpha_t \rho_t \delta_t \nabla_\theta V_\theta(s)$ for IS ratio $\rho_t \overset{\text{def}}{=} \frac{\pi(A_t|S_t)}{\mu(A_t|S_t)}$. Given state $S_t = s$, if $\mu(a|s) > 0$ when $\pi(a|s) > 0$, then the expected value of these two updates are equal. To see why, notice that

$$\mathbb{E}_\mu \left[ \alpha_t \rho_t \delta_t \nabla_\theta V_\theta(s) | S_t = s \right] = \alpha_t \nabla_\theta V_\theta(s) \mathbb{E}_\mu \left[ \rho_t \delta_t | S_t = s \right]$$

and we have

$$\begin{aligned}
\mathbb{E}_\mu \left[ \rho_t \delta_t | S_t = s \right] &= \sum_{a \in \mathcal{A}} \mu(a|s) \mathbb{E} \left[ \rho_t \delta_t | S_t = s, A_t = a \right] &&= \sum_{a \in \mathcal{A}} \mu(a|s) \frac{\pi(a|s)}{\mu(a|s)} \mathbb{E} \left[ \delta_t | S_t = s, A_t = a \right] \\
&= \sum_{a \in \mathcal{A}} \pi(a|s) \mathbb{E} \left[ \delta_t | S_t = s, A_t = a \right] &&= \mathbb{E}_\pi \left[ \delta_t | S_t = s \right].
\end{aligned}$$

Other on-policy updates can also be modified with IS ratios to adjust these action probabilities.

Though unbiased, IS can be high-variance, and so weighted IS ratios are typically preferred. For a batch consisting of transitions $\{(s_i, a_i, s_{i+1}, c_{i+1}, \rho_i)\}_{i=1}^n$, batch WIS uses a normalized estimate for the update. For example, an offline batch WIS TD algorithm would use update $\alpha_t \frac{\rho_t \delta_t \nabla_\theta V_\theta(s)}{\sum_{i=1}^n \rho_i}$. When learning online, an efficient WIS update is less straightforward, and has resulted in algorithms specialized to the tabular setting (Precup et al., 2001) or linear functions (Mahmood et al., 2014; Mahmood and Sutton, 2015). We nonetheless use WIS as baseline, in the experiments and theory.

## 3   RESAMPLING STRATEGIES FOR OFF-POLICY POLICY EVALUATION

In this section, we introduce resampling, as an alternative to importance sampling for off-policy learning. We first introduce the algorithm, Importance Resampling (IR). We then prove consistency and characterize the bias. We conclude with a discussion about its variance properties.

Importance Resampling requires access to a buffer of samples, from which we can resample. Replaying experience from a buffer was introduced as a biologically plausible mechanism to reuse old experience (Lin, 1992; 1993), and has since become common for improving sample efficiency, particularly for control (Mnih et al., 2015; Schaul et al., 2015b). In the simplest case—which we assume here—the buffer is a sliding window of the most recent $n$ samples, $\{(s_i, a_i, s_{i+1}, c_{i+1}, \rho_i)\}_{i=t-n}^t$, at time step $t > n$. These samples are generated by taking actions according to behaviour $\mu$, and so the tuples are generated with probability $d_\mu(s)\mu(a|s)\mathrm{P}(s'|s, a)$, where $d_\mu : \mathcal{S} \to [0, 1]$ is the stationary distribution for policy $\mu$. The goal is to obtain samples instead according to $d_\mu(s)\pi(a|s)\mathrm{P}(s'|s, a)$, as if we had taken actions according to policy $\pi$ from state $s \sim d_\mu$. This assumption—that states are still sampled from $d_\mu$—underlies most off-policy learning algorithms; very few attempt to use IS to adjust probabilities $d_\mu$ to $d_\pi$ (Precup et al., 2001).

The IR algorithm is simple: resample a mini-batch of size $k$ on each step $t$ from the buffer of size $n$, proportionally to $\rho_i$ in the buffer. Standard on-policy updates, such as on-policy TD or on-policy gradient TD, are then used on this resample. The key difference to IS and WIS is that the sampling distribution itself is corrected (see Theorem C.1), before the update, whereas IS and WIS correct the update itself. This small difference, however, can have larger ramifications practically, particularly with updates that accumulate in the parameters.

We consider two variants of IR: with and without bias correction. For point $t_j$ sampled from the buffer, let $\Delta_{t_j}$ be the on-policy update for that transition. For example, for TD, $\Delta_{t_j} = \delta_{t_j} \nabla_\theta V_\theta(s_{t_j})$. The first step for either variant is to sample a mini-batch of size $k$ from the buffer, proportionally to $\rho_i$, as described above. The standard IR update simply uses a mini-batch, whereas Bias-Corrected IR (BC-IR) pre-multiplies with the average ratio in the buffer $\bar{\rho}_t = \frac{1}{n} \sum_{i=1}^n \rho_i$,

$$\textbf{IR:} \qquad \alpha_t \frac{1}{k} \sum_{j=1}^k \Delta_{t_j} \qquad\qquad\qquad \textbf{BC-IR:} \qquad \alpha_t \bar{\rho}_t \frac{1}{k} \sum_{j=1}^k \Delta_{t_j}.$$

BC-IR negates bias introduced by the average ratio in the buffer deviating significantly from the true mean. For reasonably large buffers, $\bar{\rho}_t$ will be close to 1 making IR and BC-IR have near-identical updates. In practice, we find the two variants of IR perform similarly. Nonetheless, they do have different theoretical properties, particularly for small buffer sizes $n$, so we characterize both. Though BC-IR has better bias properties, IR is simpler—not requiring any modification to the updates—which may be more important than the small amount of bias introduced by the IR without bias correction. For this reason, we advocate for and analyze both variants.

Across all results, we make the following assumption.

**Assumption 1.** *Transition tuples $X_i = (S_i, A_i, S_{i+1})$ are sampled i.i.d. according to the distribution $p(x = (s, a, s')) = d_\mu(s)\mu(a|s)\mathrm{P}(s'|s, a)$, for $i = 1, 2, 3, \ldots$.*

To distinguish expectations under $p(x) = d_\mu(s)\mu(a|s)\mathrm{P}(s'|s, a)$ and $q(x) = d_\mu(s)\pi(a|s)\mathrm{P}(s'|s, a)$, we overload the notation from above, using operators $\mathbb{E}_\mu$ and $\mathbb{E}_\pi$ respectively. To reduce clutter, we will typically write $\mathbb{E}$ to mean $\mathbb{E}_\mu$, because most expectations are under the sampling distribution.

### 3.1 BIAS OF IR

We first show that IR is biased, and that its bias is actually equal to batch WIS (in Theorem 3.1). This bias is small for reasonably large $n$, because it is proportional to $1/n$. In terms of mean-squared error, composed of squared bias and variance, this term is $1/n^2$ and is relatively negligible compared to the variance. Nonetheless, for smaller buffers, such bias could have an impact. We show that with a simple modification, in BC-IR, we obtain an unbiased estimate of the update (Corollary 3.1.1).

**Theorem 3.1.** *[Bias for a fixed buffer of size $n$] Assume a buffer $B$ of $n$ transitions is sampled i.i.d., according to $d_\mu(s)\mu(a|s)\mathrm{P}(s'|s, a)$. Let $X_{\mathrm{IR}} \stackrel{\text{def}}{=} \frac{1}{k}\sum_{j=1}^{k}\Delta_{i_j}$ for transitions $i_1, \ldots, i_k$ sampled randomly from the buffer proportionally to $\rho_i$. Let $X_{\mathrm{WIS}*} \stackrel{\text{def}}{=} \sum_{i=1}^{n}\frac{\rho_i}{\sum_{j=1}^{n}\rho_j}\Delta_i$ be the batch WIS estimator of the update, with $\rho_i \stackrel{\text{def}}{=} \pi(A_i|S_i)/\mu(A_i|S_i)$. Then, $\mathbb{E}[X_{\mathrm{IR}}] = \mathbb{E}[X_{\mathrm{WIS}*}]$, and so the bias of $X_{\mathrm{IR}}$ is proportional to*

$$\mathrm{Bias}(X_{\mathrm{IR}}) = \mathbb{E}[X_{\mathrm{IR}}] - \mathbb{E}_\pi[\Delta] \propto \frac{1}{n}(\mathbb{E}_\pi[\Delta]\sigma_\rho^2 - \sigma_{\rho,\Delta}\sigma_\rho\sigma_\Delta)$$

*where $\mathbb{E}_\pi[\Delta]$ is the expected update across all transitions, with actions from $S$ taken by the target policy $\pi$; $\sigma_\rho^2 = \mathrm{Var}(\frac{1}{n}\sum_{j=1}^{n}\rho_j)$; $\sigma_\Delta^2 = \mathrm{Var}(\frac{1}{n}\sum_{i=1}^{n}\rho_i\Delta_i)$; and covariance $\sigma_{(\rho,\Delta)} = \mathrm{Cov}(\frac{1}{n}\sum_{j=1}^{n}\rho_j, \frac{1}{n}\sum_{i=1}^{n}\rho_i\Delta_i)$.*

*Proof.* Notice first that when we weight with $\rho_i$, this is equivalent to weighting with $\frac{d_\mu(S_i)\pi(A_i|S_i)\mathrm{P}(S_{i+1}|S_i,A_i)}{d_\mu(S_i)\mu(A_i|S_i)\mathrm{P}(S_{i+1}|S_i,A_i)}$, and so we are applying the correct IS ratio for the transition.

$$\mathbb{E}[X_{\mathrm{IR}}] = \mathbb{E}\left[\mathbb{E}[X_{\mathrm{IR}}|B]\right] = \mathbb{E}\left[\mathbb{E}\left[\frac{1}{k}\sum_{j=1}^{k}\Delta_{i_j}\Big|B\right]\right] = \mathbb{E}\left[\frac{1}{k}\sum_{j=1}^{k}\mathbb{E}[\Delta_{i_j}|B]\right]$$

$$= \mathbb{E}\left[\sum_{i=1}^{n}\frac{\rho_i}{\sum_{j=1}^{n}\rho_j}\Delta_i\right] = \mathbb{E}[X_{\mathrm{WIS}*}] \quad \triangleright \text{ because } \mathbb{E}[\Delta_{i_j}|B] = \sum_{i=1}^{n}\frac{\rho_i}{\sum_{j=1}^{n}\rho_j}\Delta_i$$

This bias of $X_{\mathrm{IR}}$ is therefore the same as batch WIS, which is characterized in (Owen, 2013, Section 2.7), completing the proof. □

This bias of IR will be small for reasonably large $n$, both because it is proportional to $1/n$ and because larger $n$ will result in lower variance of the average ratios and average update for the buffer. In particular, as $n$ grows, these variances decay proportionally to $n$.

**Corollary 3.1.1.** *Bias-corrected IR, with estimator $X_{\mathrm{BC}} \stackrel{\text{def}}{=} \frac{\bar{\rho}}{k}\sum_{j=1}^{k}\Delta_{i_j}$ for $\bar{\rho} = \frac{1}{n}\sum_{j=1}^{n}\rho_j$, is unbiased: $\mathbb{E}[X_{\mathrm{BC}}] = \mathbb{E}_\pi[\Delta]$.*

*Proof.*

$$\mathbb{E}[X_{\mathrm{BC}}] = \mathbb{E}\left[\frac{\bar{\rho}}{k}\sum_{j=1}^{k}\mathbb{E}[\Delta_{i_j}|B]\right] = \mathbb{E}\left[\bar{\rho}\sum_{i=1}^{n}\frac{\rho_i}{\sum_{j=1}^{n}\rho_j}\Delta_i\right] = \mathbb{E}\left[\frac{1}{n}\sum_{i=1}^{n}\rho_i\Delta_i\right] = \frac{1}{n}\sum_{i=1}^{n}\mathbb{E}\left[\frac{\pi(A_i|S_i)}{\mu(A_i|S_i)}\Delta_i\right]$$

$$= \frac{1}{n}\sum_{i=1}^{n}\mathbb{E}\left[\frac{d_\mu(S_i)\pi(A_i|S_i)\mathrm{P}(S_{i+1}|S_i, A_i)}{d_\mu(S_i)\mu(A_i|S_i)\mathrm{P}(S_{i+1}|S_i, A_i)}\Delta_i\right] = \frac{1}{n}\sum_{i=1}^{n}\mathbb{E}_\pi[\Delta] = \mathbb{E}_\pi[\Delta].$$

□

### 3.2 CONSISTENCY OF IR

Consistency of IR in terms of an increasing buffer, with $n \to \infty$, is a relatively straightforward extension on results for SIR, with or without bias correction (see Theorem C.1 in Appendix C). More interesting is consistency in terms of increasing interactions with the environment, $t \to \infty$, with a fixed length buffer, as will be the case in practice. IR, without bias correction, is asymptotically biased in this case; in fact, its asymptotic bias is the one characterized above for a fixed length buffer in Theorem 3.1. This asymptotic bias, though, is proportional to $1/n$, which is negligible for typical buffer sizes. BC-IR, on the other hand, is consistent, even with a sliding window, as we show in the following theorem.

**Theorem 3.2.** *Let $B_i = \{X_{i-n+1}, ..., X_i\}$ be the buffer of the most recent $n$ transitions sampled by time $i$, i.i.d. as specified in Assumption 1. Let $X_{\mathrm{BC}}^{(i)}$ be the bias-corrected IR estimator, with $k$ samples from buffer $B_i$. Define the sliding-window estimator $X_t \overset{\text{def}}{=} \frac{1}{t}\sum_{i=1}^{t} X_{\mathrm{BC}}^{(i)}$. Assume there exists a $c > 0$ such that $\mathrm{Var}(X_{\mathrm{BC}}^{(i)}) \leq c \; \forall i$. Then, as $t \to \infty$, $X_t$ converges in probability to $\mathbb{E}_\pi[\Delta]$.*

*Proof.* Notice first that $X_{\mathrm{BC}}^{(i)}$ is random because $B_i$ is random and because transitions are sampled from $B_i$. Therefore, given $B_i$, $X_{\mathrm{BC}}^{(i)}$ is independent of other random variables $B_j$ and $X_{\mathrm{BC}}^{(j)}$ for $j \neq i$. Now, using Corollary 3.1.1, we can show that BC-IR is unbiased for a sliding window

$$\mathbb{E}[X_t] = \mathbb{E}\Big[\frac{1}{t}\sum_{i=1}^{t} X_{\mathrm{BC}}^{(i)}\Big] = \frac{1}{t}\sum_{i=1}^{t}\mathbb{E}[\mathbb{E}[X_{\mathrm{BC}}^{(i)}|B_i]] = \mathbb{E}_\pi[\Delta].$$

Next, we show that $\lim_{|i-j| \to \infty} \mathrm{Cov}(X_{\mathrm{BC}}^{(i)}, X_{\mathrm{BC}}^{(j)}) = 0$. For $|i - j| \geq m$, $B_i$ and $B_j$ are independent, because they are disjoint sets of i.i.d. random variables. Correspondingly, $X_{\mathrm{BC}}^{(i)}$ is independent of $X_{\mathrm{BC}}^{(j)}$. Explicitly, using the law of total covariance, we get that $\mathrm{Cov}(X_{\mathrm{BC}}^{(i)}, X_{\mathrm{BC}}^{(j)}) = \mathrm{Cov}(X_{\mathrm{BC}}^{(i)}, X_{\mathrm{BC}}^{(j)}|B_i, B_j) + \mathrm{Cov}(\mathbb{E}[X_{\mathrm{BC}}^{(i)}|B_i], \mathbb{E}[X_{\mathrm{BC}}^{(j)}|B_j]) = 0$. The first term is zero because $X_{\mathrm{BC}}^{(i)}$ is independent of $X_{\mathrm{BC}}^{(j)}$ given $B_i$, and the second term is zero because $B_i$ and $B_j$ are independent. Therefore, $\lim_{|i-j| \to \infty} \mathrm{Cov}(X_{\mathrm{BC}}^{(i)}, X_{\mathrm{BC}}^{(j)}) = 0$.

Using the assumption on the variance, we can apply Lemma C.3 to $X_t$ to get the desired result. $\qquad\square$

### 3.3 VARIANCE AND EFFECTIVE SAMPLE SIZE

In this section, we provide some intuition on the variance properties of the discussed off-policy estimators. Similarly to bias, we can characterize the variance of the IR estimator relative to batch WIS. $X_{\mathrm{WIS}*}$ is able to use a batch update on all the data in the buffer, which should result in a low-variance estimate but is an unrealistic algorithm to use in practice. Instead, it provides a benchmark, where the goal is to obtain similar variance to $X_{\mathrm{WIS}*}$, but within realistic computational restrictions. Because of the clear relationship between IR and WIS, as used in Theorem 3.1, we can easily characterize the variance of $X_{\mathrm{IR}}$ relative to $X_{\mathrm{WIS}*}$ using the law of total covariance:

$$\mathbb{V}(X_{\mathrm{IR}}) = \mathbb{V}\left[\mathbb{E}[X_{\mathrm{IR}}|B]\right] + \mathbb{E}\left[\mathbb{V}[X_{\mathrm{IR}}|B]\right]$$
$$= \mathbb{V}\left[X_{\mathrm{WIS}*}\right] + \mathbb{E}\left[\mathbb{V}[X_{\mathrm{IR}}|B]\right]$$

where the variability is due to having randomly sampled buffers $B$ and random sampling from $B$. The second term corresponds to the noise introduced by sampling a mini-batch of $k$ transitions from the buffer $B$, instead of using the entire buffer like WIS. For more insight, we can expand this second term, $\mathbb{E}\left[\mathbb{V}[X_{\mathrm{IR}}|B]\right] = \mathbb{E}\left[(\frac{1}{k}\sum_{j=1}^{k}\Delta_{i_j} - \frac{1}{n}\sum_{i=1}^{n}\Delta_i)^2|B\right]$, where we consider the variance independently for each element of $\Delta_i$ and so apply the square element-wise. The variability is not due to IS ratios, and instead arises from variability in the updates themselves. Therefore, the variance of IR corresponds to the variance of WIS, with some additional variance due to this variability around the average update in the buffer.

This variance contrasts the variance of the corresponding mini-batch IS estimator:

$$\mathbb{V}\Big(\tfrac{1}{k}\sum_{j=1}^{k}\rho_{i_j}\Delta_{i_j}\Big) = \mathbb{E}\Big[\Big(\tfrac{1}{k}\sum_{j=1}^{k}\rho_{i_j}\Delta_{i_j} - \mathbb{E}_{\pi}[\Delta]\Big)^2\Big]$$

For large importance sampling ratios, this mini-batch can deviate significantly around the mean update. Further, this deviation around the mean update is across all transitions, unlike $\mathbb{E}\left[\mathbb{V}[X_{\mathrm{IR}}|B]\right]$ which reflects the expected deviation for a buffer of size $n$. The IS estimator is unbiased, as opposed to IR and WIS which both introduce some bias. A more fair comparison is to BC-IR, which is unbiased, with variance

$$\mathbb{V}\Big(X_{\mathrm{BC}}\Big) = \mathbb{E}\Big[\Big(\tfrac{\bar{\rho}}{k}\sum_{j=1}^{k}\Delta_{i_j} - \mathbb{E}_{\pi}[\Delta]\Big)^2\Big]$$

It is difficult to state generally that the variability of the IS estimator will be greater than $X_{\mathrm{BC}}$, because it depends on the properties of the update vector $\Delta_{i_j}$ itself. However, the key difference between them is that $X_{\mathrm{BC}}$ multiplies all the updates by the averaged ratio, whereas IS includes individual (potentially high magnitude) ratios inside the sum.

Finally, the variance of IR will be affected by what is known as the *effective sample size*. For data with several high magnitude ratios, and many small ratios, the IS estimator will likely suffer from high-variance updates. IR, however, will not be completely robust to this setting either: it will prevent high magnitude updates, but will be sampling from an effectively smaller dataset. This effective size is smaller because IR will repeatedly sample the same transitions, and potentially never sample some of the transitions with small IS ratios. With less data, we typically incur more variance.

One common estimator of effective sample size is $\frac{\left(\sum_{i=1}^{n}\rho_i\right)^2}{\sum_{i=1}^{n}\rho_i^2}$ (Kong et al., 1994; Martino et al., 2017). This estimate lies between $1$ and $n$. When the effective sample size is low, this indicates that most of the probability is concentrated on a few samples, which could be problematic. An important next step is to better understand theoretically the implications of effective sample size. Here, we now turn to experiments to gain more insight into the relative efficacy of IR to IS, in settings including such skewed ratios.

## 4 EXPERIMENTS

In this section, we present results for predictions made with difficult polices in three domains. We compare IR and BC-IR [2] to an importance sampling approach using uniformly sampled experiences from the replay buffer (ER+IS), and three variants of weighted importance sampling (WIS). The three variants of WIS considered are WIS-Batch, WIS-Buffer, and WIS-Optimal discussed further in the appendix[3][4]. For tabular domains, we don't average over the mini-batch updates, although doing so won't change results significantly. We report parameter studies and learning curves, excluding buffer size studies, on a single buffer size shared between all the methods. Although results from other buffer sizes (not shown here) have similar conclusions. We show $95\%$ confidence intervals on all results unless otherwise specified.

### 4.1 SETTINGS

**Random Walk Markov Chain** We use a typically constructed random walk Markov chain (Sutton and Barto, 2018) with 8 non-terminating states and 2 terminating, with a reward of 1 on the transition to the right-most terminal state and 0 everywhere else. The agent follows a policy $\mu$ and learns the value function according to a target policy $\pi$. We compute the root mean squared value error (RMSVE) on every training step with a value function found using dynamic programming with threshold $10^{-15}$.

---

[2] While BC-IR is included in the results and used for the simulated car experiments, we found IR and BC-IR to perform almost exactly the same with BC-IR having lower sensitivity to learning rate in some instances.

[3] Due to computational complexity we only test WIS-Optimal in the markov chain.

[4] We forgo evaluating any variants of the WIS update in the Torcs race car simulator based on the performance in the simpler domains.

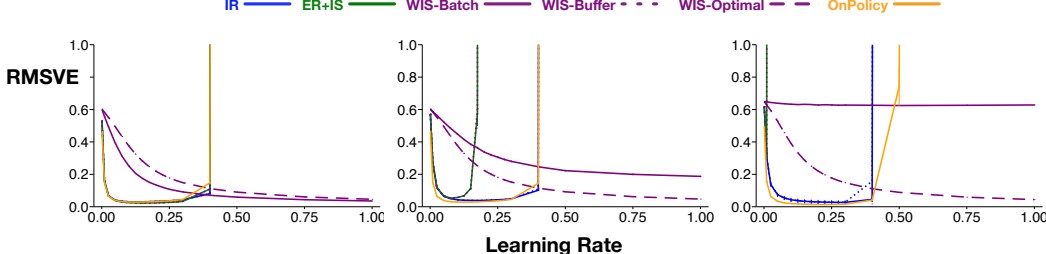

Figure 1: Learning rate sensitivity study in the Random Walk Markov Chain with buffer size $n = 15000$, batch size $b = 16$ For simplicity we write the policies as follows: $\mu = [\mu(\text{left}|\cdot), \mu(\text{right}|\cdot)]$. **left** $\mu = [0.5, 0.5], \pi = [0.1, 0.9]$, **center** $\mu = [0.9, 0.1], \pi = [0.1, 0.9]$, **right** $\mu = [0.99, 0.01], \pi = [0.01, 0.99]$. Additional results using V-Trace and Sarsa can be found in the appendix B.1

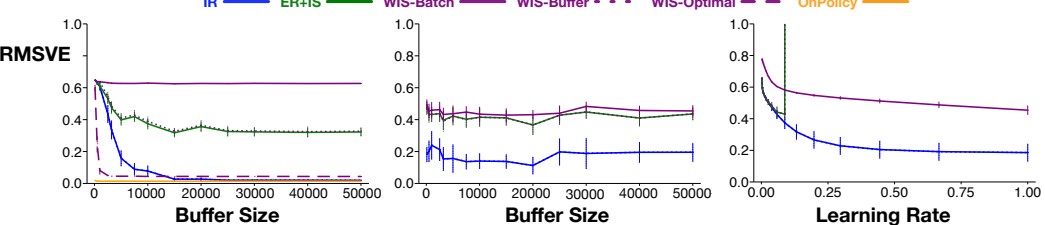

Figure 2: **left and center** Buffer size study for the random walk markov chain and four rooms domain respectively. We select the best settings for each buffersize and report the average RMSVE **right** Four rooms domain learning rate sensitivities parameter study.

**Four Rooms Environment** The four rooms domain is a well known hard domain used for training options (Stolle and Precup, 2002). The behavior policy followed by the agent is equiprobable everywhere except for 25 randomly selected states which take the action down with probability 0.05 with remaining probability split equally amongst the other actions. The 25 random states are the same for all runs, but we also evaluated different states for each run (see appendix). The target policy of interest is to take the down action deterministically, inducing highly variant IS ratios. The cumulant for the value function is 1 when the agent hits a wall and 0 otherwise. The continuation function is $\gamma = 0.9$ terminating when the agent hits a wall. We calculate the RMSVE similarly to the Markov Chain.

**Simulated Car Domain** We use the TORCs race car simulator to perform scaling experiments using neural networks. We set up the simulator to produce 64x128 cropped grayscale images. We have an underlying deterministic steering controller that produces steering actions $a_{det} \in [-1, +1]$ and take an action with probability defined by a Gaussian $a \mathcal{N}(a_{det}, 0.1)$. We show a demonstration learning a single GVF with cumulant 1 when the car is near the center of the road (signal provided by Torcs), continuation function with 0.9 everywhere with terminating condition the same as the cumulant, and a target policy modeled as a gaussian $\mathcal{N}(0.15, 0.0075)$, which corresponds to steering left.

## 4.2 RESULTS

In each of the domains tested we see significant improvements in the range of effective learning rates. The most stark example is seen in the right plot of figure 1. Here the behaviour policy induces importance sampling ratios as high as 99 while also having very few effective samples from which to train. After 1000 training epochs only a single learning rate seemed to not diverge using ER+IS and WIS-Buffer. In this same setting, IR performs very similar to the value function trained with on policy data. We also see lower sensitivity to learning rate in the four rooms environment, rightmost figure 2. [5]

Another benefit of IR is the gains in sample efficiency, and the focus on potentially rare samples following the target policy. We show this with the learning curves found in figure 3. In the random

---

[5]Additional results in the Mountain Car Domain using a three layer neural network can be found in the appendix.

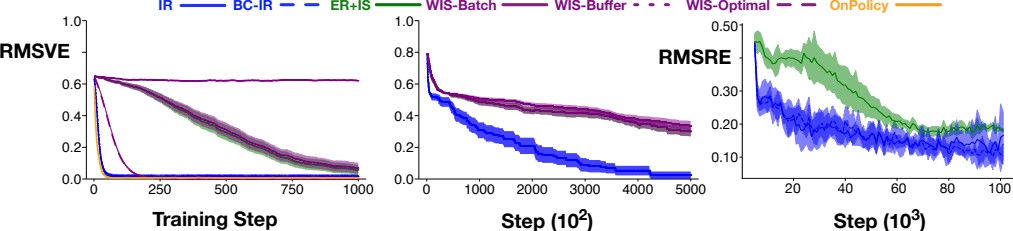

Figure 3: **left** Markov Chain Random Walk with $n = 15000$, $b = 16$, $\mu = [0.99, 0.01]$, $\pi = [0.01, 0.99]$, optimal learning rates from above parameter study **center** Four rooms environment $n = 10000$, $b = 32$ with optimal settings from below parameter studies. We report $70\%$ confidence bands for ease of comparison. **right** Torcs racing car simulator learning curves of RMSRE calculated over a pre-collected evaluation set. $\alpha_{\text{BC-IR}} = 1e^{-4}, \alpha_{\text{IR}} = 1e^{-4}, \alpha_{\text{IS}} = 1e^{-6}$ selected from a parameter study using RMSProp and the NVIDIA network designed for self-driving cars (Bojarski et al., 2016)

walk and four rooms domains we see WIS-Buffer and WIS-Batch perform approximately the same as ER+IS with IR outperforming the competitors. This can be attributed to IR's resampling scheme, where we see more samples important to training. This is especially apparent in the four rooms experiments, where we may only get a few chances to train from certain hard to reach states. The uniform sampling methods will more likely miss out on rare examples, or only see them once making learning slow. We also note that WIS-Batch is unable to learn in the hardest of settings for the Markov chain, potentially due to the bias incurred from only performing WIS on a subsample of the entire data. One interesting observation is in the Baird's Star Problem where IR results in better weights and lower value error (see appendix).

The experiments in the Torcs domain show faster learning using a deep learning system. While the results are promising, they are not as stark as we would have expected given the prior experiments and the variance in final performance is much larger for BC-IR and IR. There are several contributing factors to learning off-policy using IR and IS that need to be considered. First by using RMSProp or other adaptive learning rate algorithms we are potentially gaining WIS like benefits, reducing the variance of the updates considerably. IR may improve performance when the behaviour and target policies cause even more variant importance ratios, but with an adaptive learning rate algorithm this becomes less problematic when using IS. More needs to be understood about the interactions between adaptive algorithms and off-policy learning with importance sampling ratios.

## 5 CONCLUSIONS

Resampling for off-policy learning has been unconsidered, to our knowledge, up until now. This may be due to a focus on learning from only the most recent experiences and throwing away transitions once used. The resampling approach is now viable because of the increased prominence of the experience replay buffer in deep reinforcement learning. Previous approaches have exploited the experience replay buffer for orthogonal purposes including Prioritized Experience Replay (Schaul et al., 2015b), which prioritizes training examples according to the temporal difference error $w_i = |\delta_i| + \epsilon$. A possible extension to IR is to sample from an intermediate sampling distribution which eases high variant importance sampling ratios (see Appendix B.5).

In this paper we introduced a new approach to off-policy learning: resampling. We explored the theoretical implications of importance resampling, including a correction term to guarantee consistency in the moving window setting. We provided a number of empirical studies in hard off-policy learning settings outperforming the realistic competitors often by a wide margin, while also being less sensitive to learning rate in the mini-batch stochastic gradient update. We found IR to outperform IS in cases with potentially high importance sampling ratios suggesting a possible reduction in variance. Finally, we show improvements in learning rate performance for IR and BC-IR methods using deep learning within a challenging racing car simulator environment. There remains a number of both theoretical and empirical questions surrounding the benefits of the resampling approach to off-policy learning worth exploring in future work.

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

## A   WEIGHTED IMPORTANCE SAMPLING

We consider three weighted importance sampling updates as competitors to IR. $N$ is the size of the experience replay buffer, $b$ is the size of a single batch.

$$\Delta\theta = \frac{\sum_i^b \rho_i \delta_i \nabla_\theta V(s_i; \theta)}{\sum_j^b \rho_j} \qquad \text{WIS-Batch}$$

$$\Delta\theta = N \frac{\sum_i^b \rho_i \delta_i \nabla_\theta V(s_i; \theta)}{\sum_j^N \rho_j} \qquad \text{WIS-Buffer}$$

$$\Delta\theta = \frac{\sum_i^N \rho_i \delta_i \nabla_\theta V(s_i; \theta)}{\sum_j^N \rho_j} \qquad \text{WIS-Optimal}$$

## B   MORE EXPERIMENTAL RESULTS

### B.1   MARKOV CHAIN

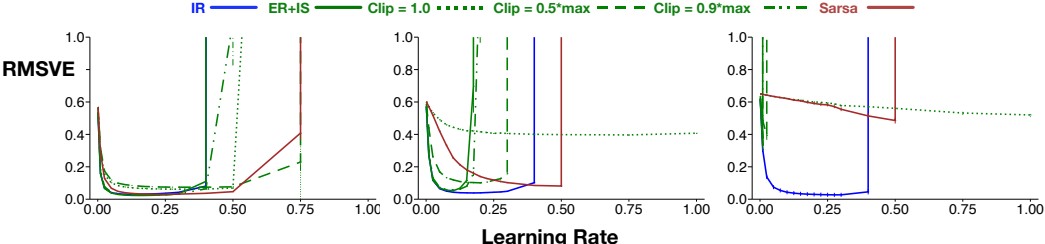

Figure 4: Markov Chain results for V-Trace and Sarsa. Four clipping parameters were chosen with different amounts of aggressiveness. They were calculated from the known max importance sampling value, multiplied by the scalars 0.5 and 0.9. We also included a clipping value of 1.0 which is consistent with the recommendations made for retrace. The error for Sarsa was calculated by deriving the state-value function from the current action-state value function, and following the same error calculation as with the other methods.

We also include results in the random walk markov chain for V-Trace and Sarsa. V-Trace exhibited the same issues as importance sampling, where we can see a clear disadvantage in the harder policies. Also, we can clearly see a bias-variance trade off in V-Trace as the clipping parameter $\bar{\rho}$ became more aggressive. Sarsa makes clear gains over ER+IS (and IR in the easiest setting), but fails to learn the true value function for the hardest settings.

### B.2   BAIRD'S STAR PROBLEM

We use a well known variant of the Star Problem (Baird, 1995; Sutton et al., 2009) proposed as a counter example to the convergence of semi-gradient temporal difference learning in the off-policy setting. We train using a batch version of TDC (Sutton et al., 2009). We calculate the RMSVE for each state on every training step.

### B.3   FOUR ROOMS DOMAIN

We also sampled new random states in which to act unfavorably for every run. We found the behavior used in the experiments presented in the main text to be harder than many of the other policies sampled randomly.

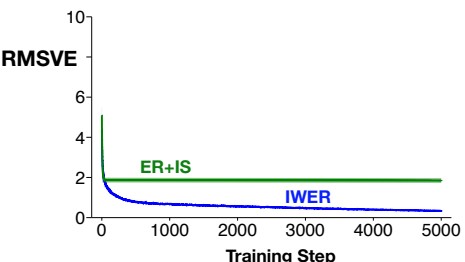

Figure 5: Baird's Star Problem.

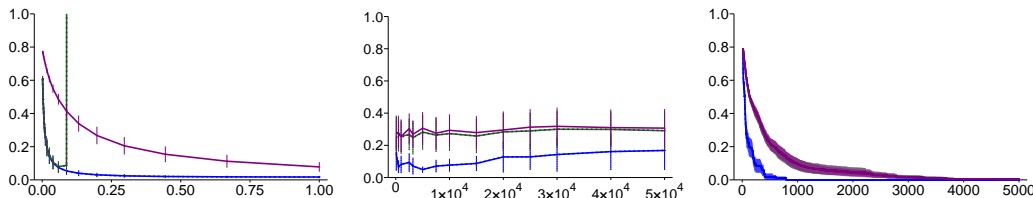

Figure 6: Four rooms experiments for new random states every run: **left** Learning rate sensitivity **center** Buffer Size Sensitivity **right** Learning Curves

## B.4  MOUNTAIN CAR

We use the standard mountain car domain described in (Sutton and Barto, 2018). To make the simulations more realistic we collect experience from behavior policy learned through Q-learning (Sutton and Barto, 2018) with a $\epsilon$-greedy exploration strategy. We code a GVF to predict whether the agent will hit the back wall within a horizon of $\gamma = 0.9$ while following the the persistent policy accelerating forward. We use two exploration parameters $\epsilon = \{0.1, 0.5\}$ with maximum importance sampling ratio values at $\rho_{max} = \{6, 30\}$ respectively.

We show comparisons for both a static step size SGD and with the Adam optimizer. These experiments perform as expected, except for one configuration of the behavior policy using the Adam optimizer. It is possible that the Adam optimizer is getting some benefit similar to WIS with the recency averages of updates accounting for the high variance of the update.

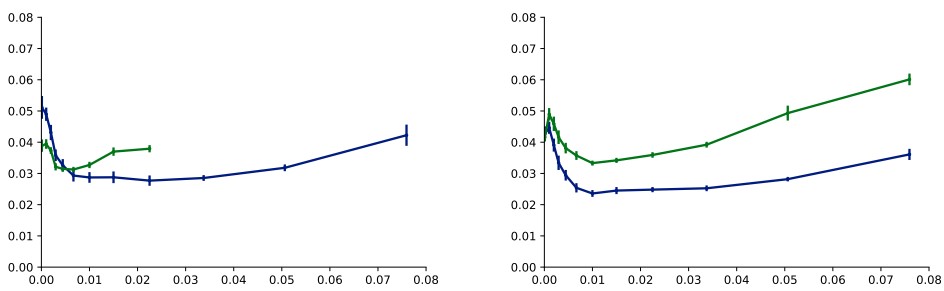

Figure 7: **Mountain Car**: Mini-batch Gradient Descent with constant learning rate $\pi = [0, 0, 1.0], |\mathcal{B}| = 50000$, **Left** $\epsilon = 0.1$, **Right** $\epsilon = 0.5$

## B.5  SAMPLING ACCORDING TO INTERMEDIATE POLICIES

While intuitively it might seem sampling according to the target policy will produce the best value functions, when making multiple predictions with the same training network it would be more convenient to sample from the experience replay buffer with an intermediate policy. We get the benefits of less variant importance sampling ratios with the ability to use one IR buffer for multiple policies.

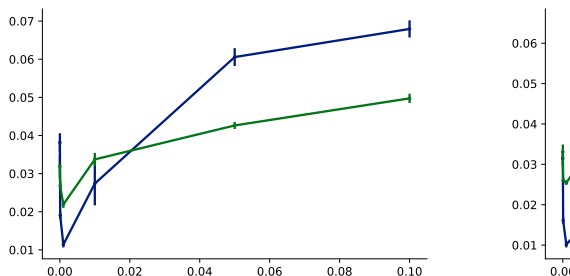
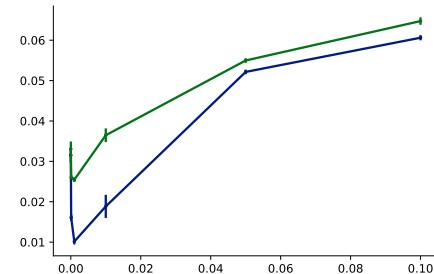

Figure 8: **Mountain Car**: Mini-batch Gradient Descent with Adam Optimizer. x-axis: learning rates, y-axis: RMSVE $\pi = [0, 0, 1.0], |\mathcal{B}| = 50000$, **Left** $\epsilon = 0.1$, **Right** $\epsilon = 0.5$

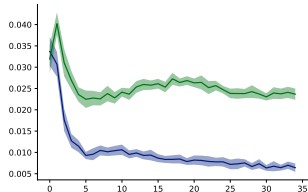
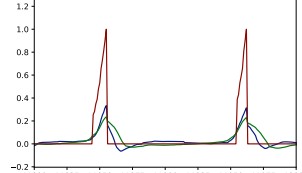
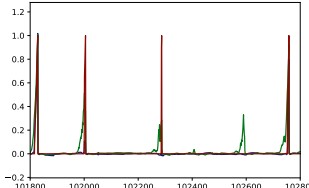

Figure 9: **Mountain Car left** Learning curve for optimal settings for Adam optimizer x-axis: time($10^4$) y-axis: RMSVE **center** Early example of learned predictions x-axis: Time y-axis: Prediction **right** Late example of learned predictions x-axis: Time, y-axis: Prediction

To combine the IR and importance sampling with a uniform experience replay we sample from the buffer according to the PMF

$$p(\text{sampling } d_i) = \frac{\bar{\rho}_i}{\sum_j \bar{\rho}_j}, \qquad \bar{\rho}_i = \frac{\pi_{sample}(a_t|s_t)}{\mu(a_t|s_t)}.$$

We would then us off-policy temporal difference methods with the effective importance sampling ratio used as $\rho_{\text{effective}} = \frac{\pi(a_t|s_t)}{\pi_{sample}(a_t|s_t)}$. We show initial results for this extension in the markov chain random walk in figure 10.

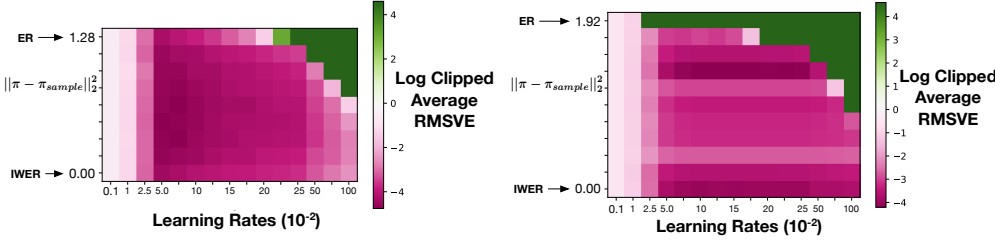

Figure 10: Intermediate sampling policies in a random walk markov chain. (left) $\pi = [0.1, 0.9], \mu = [0.9, 0.1]$, (right) $\pi = [0.01, 0.99], \mu = [0.99, 0.01]$

## C   MORE THEORETICAL RESULTS

### C.1   CONSISTENCY OF THE IR ESTIMATOR WITH GROWING BUFFER SIZE

We show the consistency of the IR estimator with $n \to \infty$ for convenience, but our approach closely follows that of (Smith et al., 1992).

**Theorem C.1.** *Let $B = \{x_1, x_2, ..., x_n\}$ be a buffer of data sampled i.i.d. according to proposal distribution $p(x)$. Let $q(x)$ be some distribution of interest and assume the proposal distribution samples everywhere where $q(x)$ is non-zero. Also, let $Y$ be a discrete random variable taking values $x_i$ with probability $\propto \frac{q(x_i)}{p(x_i)}$.*
*Then, $Y$ converges in distribution to $X \sim Q$ as $n \to \infty$.*

*Proof.* Let $\rho_i = \frac{q(x_i)}{p(x_i)}$. From the probability mass function of $Y$, we have that:

$$\mathbb{P}[Y \leq a] = \sum_{i=1}^{n} \mathbb{P}[Y = x_i] \mathbb{1}\{x_i \leq a\}$$

$$= \frac{n^{-1} \sum_{i=1}^{n} \rho_i \mathbb{1}\{x_i \leq a\}}{n^{-1} \sum_{i=1}^{n} \rho_i}$$

$$\xrightarrow{n \to \infty} \frac{\mathbb{E}_q[\rho(x) \mathbb{1}\{x \leq a\}]}{\mathbb{E}_q[\rho(x)]}$$

$$= \frac{1 \cdot \int_{-\infty}^{a} \frac{q(x)}{p(x)} p(x) dx + 0 \cdot \int_{a}^{\infty} \frac{q(x)}{p(x)} p(x) dx}{\int_{-\infty}^{\infty} \frac{q(x)}{p(x)} p(x) dx}$$

$$= \int_{-\infty}^{a} q(x) dx$$

$$\square$$

The above shows consistency. If we have a large enough buffer, then the importance resampling will closely approximate sampling from the target distribution $Q$. In particular, any expectation we want to estimate under $Q$ can also be a well-approximated by using the defined sampling scheme.

## C.2 CHANGING BEHAVIOUR POLICIES

We consider the importance sampling estimator under conditions of a changing policy.

**Theorem C.2.** *Let $P$ be a target distribution with density $p$ and $\{x_i\}_{i=1}^{n}$ be a dataset where each $x_i$ is sampled from $P_i$ independently. Then, the weighted importance sampling estimator $\hat{\mu} = \sum_{i=1}^{n} \frac{\rho_i}{\sum_{j=1}^{n} \rho_j} g(x_i)$ is consistent for $\mathbb{E}_f[g(X)]$ where $g(.)$ is a function of interest and $\rho_i = \frac{q(x_i)}{p_i(x_i)}$.*

*Proof.* First, we rewrite $\hat{\mu} = \frac{\frac{1}{n} \sum_{i=1}^{n} \rho_i g(x_i)}{\frac{1}{n} \sum_{j=1}^{n} \rho_j}$ and let $Y_i = \rho_i g(x_i)$.

Next, we find the expectations of $Y_i$ and $\rho_i$.

$$\mathbb{E}_{p_i}[Y_i] = \int \frac{q(x)}{p_i(x)} g(x) p_i(x) dx$$

$$= \mathbb{E}_q[g(X)]$$

$$\mathbb{E}_{p_i}[\rho_i] = \int \frac{q(x)}{p_i(x)} p_i(x) dx$$

$$= \int q(x) dx$$

$$= 1$$

Finally, since all $Y_i$ and all $\rho_i$ have the same expectation and are independent, we apply the law of large numbers to obtain that

$$\hat{\mu} = \frac{\frac{1}{n} \sum_{i=1}^{n} \rho_i g(x_i)}{\frac{1}{n} \sum_{j=1}^{n} \rho_j} \xrightarrow{n \to \infty} \frac{\mathbb{E}_q[g(X)]}{\quad} \qquad (1)$$

as desired.

$\square$

Theorem C.2 implies that, even if the behaviour policy is changing over time, as long as we use an importance ratio corresponding to the policy that was used to select an action, we will retain an unbiased estimate of the expected update.

## C.3 CONSISTENCY UNDER A SLIDING WINDOW DATASET

**Lemma C.3.** *Let $Z_1, ...Z_n$ be random variables with mean $\mu$. Suppose there exists a $c > 0$ such that $\mathbb{V}(Z_i) \leq c \ \forall i$ and that $\lim_{|i-j| \to \infty} \text{Cov}(X_i, X_j) = 0$.*
*Then, as $N \to \infty$, $\frac{1}{N} \sum_{i=1}^{N} Z_i$ converges in probability to $\mu$.*

*Proof.* Let $S_N = \sum_{i=1}^{N} Z_i$.

$$\mathbb{V}(S_N) = \sum_{i=1}^{N} \mathbb{V}(Z_i) + 2 \sum_{i=1}^{N} \sum_{j=i+1}^{N} \text{Cov}(Z_i, Z_j)$$

The first term is bounded by $cN$ from our assumption on the variance. Now, to bound the second term.

Fix $\delta > 0$ and choose $M$ such that $\forall |i - j| > M$, $|\text{Cov}(Z_i, Z_j)| < \delta$ (such an $M$ must exist since $\lim_{|i-j| \to \infty} \text{Cov}(X_i, X_j) = 0$). Assuming that $N > M$, we can decompose the second term into

$$\sum_{i=1}^{N} \sum_{j=i+1}^{N} \text{Cov}(Z_i, Z_j) = \sum_{i=1}^{N} \sum_{j=i+1}^{i+M} \text{Cov}(Z_i, Z_j) + \sum_{i=1}^{N} \sum_{j=i+M+1}^{N} \text{Cov}(Z_i, Z_j)$$

$$\left| \sum_{i=1}^{N} \sum_{j=i+1}^{N} \text{Cov}(Z_i, Z_j) \right| \leq \sum_{i=1}^{N} \sum_{j=i+1}^{i+M} |\text{Cov}(Z_i, Z_j)| + \sum_{i=1}^{N} \sum_{j=i+M+1}^{N} |\text{Cov}(Z_i, Z_j)|$$

By the Cauchy-Schwarz inequality and our variance assumption, $|\text{Cov}(Z_i, Z_j)| \leq c$. So, we get

$$\left| \sum_{i=1}^{N} \sum_{j=i+1}^{N} \text{Cov}(Z_i, Z_j) \right| \leq \sum_{i=1}^{N} \sum_{j=i+1}^{i+M} c + \sum_{i=1}^{N} \sum_{j=i+M+1}^{N} \delta$$
$$\leq NMc + N^2 \delta$$

Altogether, our upper bound is
$$\mathbb{V}\left(\frac{S_N}{N}\right) \leq \frac{c}{N} + \frac{Mc}{N} + \delta$$

Finally, we apply Chebyshev's inequality. For a fixed $\epsilon > 0$,

$$P\left(\left|\frac{S_N}{N} - \mu\right| > \epsilon\right) \leq \frac{1}{\epsilon^2} \mathbb{V}\left(\frac{S_N}{N}\right)$$
$$\leq \frac{1}{\epsilon^2}\left(\frac{c}{N} + \frac{Mc}{N} + \delta\right)$$

Since we can choose $\delta$ to be arbitrarily small (say $\delta = \frac{1}{N}$), the right-hand side goes to 0 as $N \to \infty$, concluding the proof.

$\square$

