# OpenReview forum: "Importance Resampling for Off-policy Policy Evaluation"
_ICLR.cc/2019/Conference_

### Official Review · AnonReviewer2 · 2018-11-05
**Interesting approach, but unclear how far it is applicable**

**Rating:** 5
**Confidence:** 3

**Review:**

The authors propose to use importance resampling (IR) in place of importance sampling (IS) for policy evaluation tasks. The method proposed by the authors definitely seems valid, but it isn’t quite clear when this is applicable.

IR is often used in the case of particle filters and other SMC is often used to combat the so-called “degeneracy problem” where a collection of particles (or trajectories) comes to degenerate such that all the mass is concentrated onto a single particle. This does not seem to be the case here, as the set of data (the replay buffer) does not seem to be changing over time. In particular, since the target policy and the behavior policy are fixed, the bigger issue seems to be that the distribution itself will not change over time.

Finally, the results are given for somewhat simple problems. The first two settings show that the difference between IR/IS can be very stark, but it seems like this is the case when the distributions are very different and hence the ESS is very low. The IR methods seem like they can eliminate this deficiency by only sampling from this limited subset, but it is also unclear how to extend this to the policy optimization setting.

Overall I have questions about where these results are applicable. And finally, as stated a moment ago, it is unclear how these results could be extended to the setting of off-policy policy optimization, where now the resulting policies are changing over time. This would necessitate updating the requisite sampling distributions as the policies change, which does seem like it would be difficult or computationally expensive (unless I am missing something). Note that this is not an issue with IS-based methods, because they can still be sampled and re-weighted upon sampling.

---

> ### Author Response · Authors · 2018-11-08
> **Author Response**
>
> Thank you for your review, and look forward to having a discussion about your comments and concerns!
>
> From reading your review you have two main concerns and these concerns are quite strongly linked: what is the applicability of this algorithm, and how can this be used for control. We focus on these two concerns in the main body and address other concerns below.
>
> While a large portion of the RL community is primarily concerned with control there is also interest in pure policy evaluation, or prediction, where our algorithm is highly applicable. Concretely, we look towards the Horde architecture [1], which is a large collection of general value functions (GVFs). This type of architecture could benefit from variance reduction techniques designed for static target policies, especially if the predictive units are using a shared representation. Our algorithm provides variance reduction, and also prioritizes samples important for learning off-policy value functions. Another application is in the autonomous car domain, as represented with the experiments in torcs, where predictions are made about certain pre-defined policies that the car cannot take due to safety concerns but can learn about off-policy.
>
> You comment that our algorithm works best when policies are very different. We agree! And find this an especially appealing property of resampling. If our goal is to learn a large collection of GVFs with a potentially diverse set of target policies we want an algorithm which can be applied no matter the behaving policy. You may also have concerns about computational complexity with using this algorithm for a large collection of GVFs with many target policies. Instead of keeping a PMF for each new target policy, we could instead keep a single PMF with a policy that is well supported for all the target distributions (i.e. a uniform random policy). We may be able to choose a sampling policy to produce a lower variant value function than the target policy (see [2] about the best proposal distribution q(x) for a statistic f(x) and p(x) target distribtuion being q(x) \propto f(x)p(x)).
>
> Applying resampling to control is possible as one could apply importance sampling to the change in the target policy new_tp(x)/old_tp(x) compared to what the target policy was when it was first stored in the replay buffer.  This is computationally efficient and should address the concerns that our method cannot be applied to the control setting efficiently.  The benefits of IR in the control setting are still to be seen but we think we should see reduced variance in the updates which could be quite beneficial. This extension would be interesting for follow up work. There is also no need to use the current target policy as the sampling policy, as mentioned above.
>
> (Continues below)

---

> > ### Author Response · Authors · 2018-11-08
> > **Author response part 2**
> >
> > Other concerns:
> >
> > The two domains in which we do more concrete empirical studies are simple, but the simplicity allows us to make concrete statements about the effects of our algorithm. One example is in the four rooms domain, where competitors could not learn given 500,000 examples from the environment, where IR performed well given the same amount of data almost learning the entire value function in unfavorable conditions. We also provide a demonstration in torcs, and theoretical contributions about the approach's bias and variance.
> >
> > We are unsure exactly what you mean by "In particular, since the target policy and the behavior policy are fixed, the bigger issue seems to be that the distribution itself will not change over time" but will respond to the best of our ability. Any clarifications for future discussion will be helpful.
> >
> > There is no need for the behaviour policy to be fixed, but we follow many of the experimental designs from prior off-policy work. We also provide examples in the appendix with a learned behaviour policy in mountain car. The changing target policy (as in the control setting) could also be handled as mentioned above.
> >
> > The replay buffer is not being completely resampled (i.e. SIR [3]). Correct. Although we are gaining the benefits of this type of full resample. We decide not to do a full resample (as in SIR), because the problem for an RL agent is a bit different. It seems better to keep the old replay buffer as is and resample only portions non-destructively so the buffer could be shared for other portions of the full autonomous agent. In a sense you could imagine us constructing a resampled buffer, but not storing the resampled buffer.
> >
> > Finally the replay buffer is changing overtime through a moving window of past experience, as is typical in reinforcement learning applications in the Torcs domain and Four rooms domain. We will work to make clear in the final version that only the Markov chain has a fixed set of experience (primarily so we can do clearer studies).
> >
> > [1] Sutton, Richard S., et al. "Horde: A scalable real-time architecture for learning knowledge from unsupervised sensorimotor interaction." The 10th International Conference on Autonomous Agents and Multiagent Systems-Volume 2. International Foundation for Autonomous Agents and Multiagent Systems, 2011.
> > [2] Art B. Owen. 2013. Monte Carlo theory, methods and examples. Chapter 9: http://statweb.stanford.edu/~owen/mc/
> > [3] Rubin, Donald B. "Using the SIR algorithm to simulate posterior distributions." Bayesian statistics 3 (1988): 395-402.

---

### Official Review · AnonReviewer4 · 2018-11-09
**Simple and interesting method; some questions with the main theoretical results; would like to see the comparison with FQI**

**Rating:** 5
**Confidence:** 3

**Review:**

This paper introduces the concept of Sampling Importance Resampling (SIR) and give a simple method to adjust the off-policyness in the TD update rule of (general) value function learning, as an alternative of importance sampling. The authors argue that this resampling technique has several advantages over IS, especially on the stability with respect to step-size if we are doing optimization based the reweighted/resampled samples. In experiment section they show the sensitivity to learning rate of IR TD learning is closer to the on-policy TD learning, comparing with using IS or WIS.

Main comments:
The proposed IR technique is simple and definitely interesting in RL settings. The advantage about sensitivity of step-size choice in optimization algorithm looks appealing to me, since that is a very common practical issue with IS weighted objective. However I feel both the theoretical analysis and empirical results will be more convinced to me if a more complete analysis is presented. Especially considering that the importance resampling itself is well known in another field, in my point of view, the main contribution/duty of this paper would be introducing it to RL, comparing the pros/cons with popular OPPE methods in RL, and characterize what is the best suitable scenario for this method. I think the paper could potentially do a better job. See detailed comments:

1. The assumption of Thm 3.2 in main body looks a little bit unnatural to me. Why can we assume that the variance is bounded instead of prove what is the upper bound of variance in terms of MDP parameters? I believe there exists an upper bound so that result would be correct, but I’m just saying that this should be part of the proof to make the theorem to be complete.
2. If my understanding to section 3.3 is correct, the variance of IR here is variance of IR just for one minibatch. Then this variance analysis also seems a little bit weird to me. Since IR/IR-BC is computed online (actually in minibatch), I think a more fair comparison with IS/WIS might be giving them the same number of computations over samples. E.g. I would like to see the result of averaged IR/IR-BC estimator (over n/k minibatch’s) in either slicing window (changed every time) or fully offline buffer, where n is the number of samples used in IS/WIS and k the size of minibatch. I think it would be more informative than just viewing WIS as an (upper bound) benchmark since it uses more samples than.
3. From a higher level, this paper considers the problem of learning policy-value function with off-policy data. I think in addition to TD learning with IS adjustment, fitted Q iteration might be a natural baseline to compare with. It is also pretty widely-used and simple. Unlink TD, FQI does not need off-policy adjustment since it learns values for each action. I think that can be a fair and necessary baseline to compare to, at least in experiment section.
4. A relatively minor issue: I’m glad to see the author shows how sensitive each method is to the change of learning rate. I think it would be better to show some results to directly support the argument in introduction -- “the magnitude of the updates will vary less”, and maybe some more visualizable results on how stable the optimization is using IS and IR. I really think that is the most appealing point of IR to me.

Minor comments:
5. The authors suggest that the second part of Var(IR), stated in the fifth line from the bottom in page 5, is some variability not related to IS ratio but just about the update value it self. I think that seems not the case since the k samples (\delta_ij’s, j=1 to k) actually (heavily) depend on IS raios, unless I missed something here. E.g. in two extreme case where IS weights are all ones or IS weights are all zero except for one (s,a) in the buffer, the variance is very different and that is because of IS ratio but not the variance of updates themselves.
6. Similar with (5), in two variance expressions on the top of page 6, it would be better to point out that the distribution of k samples are actually different in two equations. One of them is sampled uniformly from buffer and the other is proportional to IS ratios.
7. I think it is a little bit confused to readers when sometimes both off-policy learning and off-policy policy evaluation are used to describe the same setting. I would personally prefer use off-policy (policy) learning only in the “control” setting: learning the optimal policy or the optimal value function, and use the term off-policy policy evaluation referring to estimating a given policy’s value function. Though I understand that sometimes we may say “learning a policy value function for a given policy”, I think it might be better to clarify the setting and later use the same term in the whole paper.

Overall, I think there are certainly some interesting points about the IR idea in this paper. However the issues above weakens my confidence about the clarity and completeness of the analysis (in both theory and experiment) in this paper.

---

> ### Author Response · Authors · 2018-11-14
> **Author Response**
>
> Thank you for the review! Your comments were insightful. We answer below:
>
> 1.
> We can bound the variance if we assume bounds on all the individual quantities, but the specific bound would not be important since we only need the TD update to have finite variance. We could mention the specific quantities that need bounding in the final version (i.e. variance of rewards, gradients, ...).
>
> 2.
> From what we gather, the reviewer wants a more careful comparison of the variances for the same number of samples. Unfortunately, there is no simple expression for the variance of the WIS estimator. The expressions that do exist use various approximations to estimate the variance of WIS. We also don't believe this would be a useful comparison, as one is a batch (WIS) while the other is an online mini-batch (IR) algorithm. A potential better comparison would be the progress in terms of the objective function over the same number of samples. Unfortunately, we don't think this would be possible as off-policy TD has no convergence guarantees with function approximation.
>
> 3.
> Our understanding is FQI is a control algorithm, as it is defined by [1]. We don't think it is an applicable competitor for two reasons. The first is we aren't doing control here, and the algorithm was designed specifically for off-policy policy evaluation. All of our prediction tasks have set policies for which we want to evaluate, we aren't controlling to maximize a signal. Another point, we are not learning state-action value functions but only state-value functions (off-policy algorithm papers typically focus on either state value [2][3] or  state-action value [4][5]).
>
> This does not mean we are unable to extend the algorithm to the control case (see comments for reviewer 2) or state-action value functions, but decided for this paper to focus on state-value functions.
>
> 4.
> We will consider including this in the final paper or in the appendix, but feel as though the sensitivity curve is a relatively good proxy for measuring the variance of the updates. If the curve is wide this means the updates are less variant, while a narrow curve the updates are variant and small learning rates must be used. We also think the empirical reduction in variance is obvious from the removal of the IS ratio from the update (the major contributor to high variance in IS).
>
> 5 & 6.
> Right! The variance of the second term (\expected[\var(X_IR)]) is dependent on the IS ratios, because of the sampling distribution. We can clarify this and point 6 in the final version.
>
> 7.
> While we are focusing on off-policy policy evaluation, we do not feel as though off-policy learning is restricted to the control case. Instead we view off-policy learning as the more general statement of learning from off-policy data, which could be policy evaluation or policy improvement.
>
>
> [1] Ernst, Damien, Pierre Geurts, and Louis Wehenkel. "Tree-based batch mode reinforcement learning." Journal of Machine Learning Research 6.Apr (2005): 503-556.
> [2] Sutton, Richard S., et al. "Fast gradient-descent methods for temporal-difference learning with linear function approximation." Proceedings of the 26th Annual International Conference on Machine Learning. ACM, 2009.
> [3] Mahmood, A. Rupam, Hado P. van Hasselt, and Richard S. Sutton. "Weighted importance sampling for off-policy learning with linear function approximation." Advances in Neural Information Processing Systems. 2014.
> [4] Munos, Rémi, et al. "Safe and efficient off-policy reinforcement learning." Advances in Neural Information Processing Systems. 2016.
> [5] Mahmood, Ashique Rupam, Huizhen Yu, and Richard S. Sutton. "Multi-step off-policy learning without importance sampling ratios." arXiv preprint arXiv:1702.03006 (2017).

---

> > ### Comment · AnonReviewer4 · 2018-11-18
> > **Thanks for your response; some clarification about FQI**
> >
> > Thanks for your response! Just some clarification about what I meant by fitted Q:
> >
> > It is not exactly the original FQI algorithm, but I believe it is a simple variant of FQI in policy evaluation case. (Maybe it has another name?) Note that fitted Q iteration use (s,a,s') pairs from any behavior policy, to learn a (optimal) Q function by fitting the following optimality Bellman equation:
> > Q(s,a) = r(s,a) + \gamma*max_a' Q(s',a')
> >
> > If we want to do policy evaluation, we can simply change the optimality Bellman equation to policy Bellman equation:
> > Q(s,a) = r(s,a) + \gamma* E_{a' \sim \pi(s')} Q(s',a')
> > And this does not require IS ratio too, as FQI for policy optimization.

---

> > > ### Author Response · Authors · 2018-11-18
> > > **Only possible for state-action value functions**
> > >
> > > Oh right. In the state-action value function case we can do this. But I'm pretty sure this isn't possible for the state value function case, which is what is considered here.
> > >
> > > If you have evidence to the contrary, let use know! :)
> > >
> > > Thank you for the clarification!

---

> > > > ### Comment · AnonReviewer4 · 2018-11-18
> > > > **Agree, but can we compute V(s) if you already get Q(s,a)?**
> > > >
> > > > Totally agree and that why I'm thinking about FQI can be a simple baseline: TD-learning of V(s) requires IS ratio in off-policy case, but for learning Q(s,a) we may not need that, so we are not even bothered by the problem of IS-ratio in state value function learning.
> > > >
> > > > After we learn the state-action value function, we can compute the state value function since we know the policy. In that point of view, I guess it should be directly comparable in experiment?
> > > >
> > > > In general, I do agree there is many cases we need IS ratio so this method is useful and not directly comparable with FQI, e.g. we can replace the original IS (just IS itself) approach in OPPE with this IR.
> > > >
> > > > Also, one thing I might miss is, this generalized value function learning problem may be more general than FQI case where we need to assume MDP? I'm not familiar with the GVF setting described in the paper.

---

> > > > > ### Author Response · Authors · 2018-11-26
> > > > > **Please see the revision**
> > > > >
> > > > > We found your argument hard to refute, and thus performed experiments in the markov chain random walk setting in the current revision. Please review the revision and let us know if there are any further concerns.

---

### Official Review · AnonReviewer5 · 2018-11-12
**Good idea on off-policy learning, but with limited analysis and experiments**

**Rating:** 6
**Confidence:** 4

**Review:**

In this work, the authors studied the technique of importance re-sampling (IR) for off-policy evaluation in RL, which tends to have low-biased (and it's unbiased in the bias-correction version) and low-variance. Different than existing methods such as importance sampling (IS) and weighted importance sampling (WIS) which correct the distribution over policy/transitions by an importance sampling ratio, in IR one stores the offline data in a buffer and re-samples the experience data (in form of state, action, & next-state) for on-policy RL updates. This approach avoids using importance sampling ratios directly, which potentially alleviate the variance issue in TD estimates. The authors further analyze the bias and consistency of IR, discuss about the variance of IR, and demonstrate the effectiveness of IR by comparing it with IS/WIS on several benchmark domains.

On the overall I think this paper presents an interesting idea of IR for off-policy learning. In particular it hinges on designing the sampling strategy in replay buffer to handle the distribution discrepancy problem in off-policy RL. Through this simple off-policy estimator, the authors are able to show improvements when compared with other state-of-the-art off-policy methods such as IS and WIS, which are both known to have high-variance issues. The authors also provided bias and consistency analysis of these estimators, which are reasonable theoretical contributions. The major theoretical question/concern that I have is in terms the variance comparisons between IR and IS/WIS. While I see some discussions in Sec 3.3, is there a concrete result showing that IR estimator has lower variance when compared to IS and WIS (even under certain technical assumptions)? This is an important missing piece for IR, as the original motivation of not using IS/WIS estimators is because of their issues on variance.

In terms of experiment, while the authors have done a reasonably good job evaluating IR on several domains based on the MSE of policy evaluation, to make it more complete can the authors also show the efficiency of IR when compared to state-of-the-art algorithms such as V-trace, ABQ or Re-Trace (which are cited in the introduction section)?

---

> ### Author Response · Authors · 2018-11-19
> **Author Response**
>
> Thank you for your review! We will address you concerns below.
>
> You are correct, we don’t show concrete evidence of IR having lower variance than IS or WIS. This is actually quite a tricky thing to do in general, and the WIS* estimator we use as a lower bound for the variance of IR also doesn’t have a good form. Unfortunately, we are unaware of any techniques to show this more concretely but provided the bounds as presented in the paper for completeness. In fact, it is not generally true that WIS has lower variance than IS, and proving more generally that IR (or even WIS) has lower variance than IS is not possible here.
>
> We don’t compare to retrace [4] and ABQ [3] for two reasons. The first is we focus on State-values in this paper, while ABQ and retrace are only derived for state-action values. The other reason (specifically for ABQ) is that we cannot use a trace in the experience replay setting. ABQ [3] was primarily designed as an online algorithm, requiring the trace to learn off-policy. When sampling from a replay buffer, an eligibility trace doesn’t make sense as the temporal structure of the data is broken. Finally, V-trace wasn’t compared because we care about getting accurate values for the target policies as exactly specified, rather than policies in-between the target and behavior [1]. This minor point is inconsequential for the use of V-trace in IMPALA as long as the ordering of policies remains consistent (i.e. the critic is still valid), but our main goal is to (as exactly as possible) evaluate policies towards creating many GVF predictions (Horde like architecture [2]). We give some hypothesis below about how V-trace would perform, where we don’t feel it would add much outside of the comparisons already made. We felt focusing on fundamental approaches (IS, WIS) to importance resampling was fair here, and didn't feel V-trace would have added any more useful comparisons.
>
> Hypothesis for how V-trace will perform:
>
> Markov Chain Random Walk:
> Easiest policy: It will perform as well as IS, assuming the clipping parameter c_i is set well (i.e. above 2), making the algorithm equivalent to importance sampling (IS)
> hardest policy: It will be less sensitive than IS (depending on what the clipping parameter c_i is set to), but will still face the same issues as ER+IS when sampling from the experience replay buffer. Because there is no prioritization on the “important” samples, we expect the RMSVE to follow closely with ER+IS in figure 3 (left).
>
> Four Rooms:
> We expect V-trace to have similar problems to the hardest policy, similar to IS, where we get a broader range of useful learning rates but still are hampered by the experience sampled.
>
> Torcs:
> It is hard to know here. We expect V-trace to work well, but there is a lot of play in how the learning rate is tuned (RMSProp) that makes this problem hard to predict.
>
>
> [1] Espeholt, Lasse, et al. “IMPALA: Scalable distributed Deep-RL with importance weighted actor-learner architectures.” arXiv preprint arXiv:1802.01561 (2018).
> [2] Sutton, Richard S., et al. “Horde: A scalable real-time architecture for learning knowledge from unsupervised sensorimotor interaction.” The 10th International Conference on Autonomous Agents and Multiagent Systems-Volume 2. International Foundation for Autonomous Agents and Multiagent Systems, 2011.
> [3] Mahmood, Ashique Rupam, Huizhen Yu, and Richard S. Sutton. “Multi-step off-policy learning without importance sampling ratios.” arXiv preprint arXiv:1702.03006 (2017).
> [4] Munos, Rémi, et al. “Safe and efficient off-policy reinforcement learning.” Advances in Neural Information Processing Systems. 2016

---

> > ### Public Comment · (anonymous) · 2018-12-10
> > **Thank you for the clarifications**
> >
> > I have gone through the reviews and found the explanations reasonable. I also agree that theoretical comparisons between the variance of IR, IS (and/or WIS) is non-trivial, and there seems to be no known analysis on this.
> >
> > Re comparisons, although the above explanations (on ABQ, Vtrace, Impala) are very reasonable, I still think comparing with at least one modern off-policy RL methods (such as V-trace) would be useful to convince readers on the idea of applying IR to off-policy RL, especially empirically showing how this method can better utilize replay buffer in learning.
> >
> > Therefore, I would keep my current scores.

---

### Author Response · Authors · 2018-11-26
**Author Revision**

We have submitted a minor revision to the current paper, primarily including results for V-trace and Sarsa in the appendix and pointing to these results in the main paper. From these results our initial hypothesis seem correct. To recap here, V-trace does well, but there is a clear variance bias trade-off as the clipping parameter (\bar{\rho}) becomes more aggressive (also having similar issues to ER+IS in the hardest policy settings). Sarsa performed well for the first two policy settings (the easiest) in the markov chain, while not learning in the final setting. We believe Sarsa breaks down in the hardest case for similar reasons that ER+IS does (not sampling enough of the needed experience to learn). We decided to again exclude ABQ, as with the trace parameter set to 0 (which is what is considered here because of the replay buffer) the algorithm resolves down to TD(0).

From these preliminary results we don’t feel the two added algorithms (V-trace and Sarsa) add to the comparison meaningfully. We stand by keeping the current results in the paper as is, with the added algorithms tested in the appendix.

We decided to forgo changes to the theory section for the same reasons mentioned in the individual reviews.

We hope you get a chance to review the additions, and look forward to further comments you may have.

---

### Meta-Review · Area_Chair1 · 2018-12-14
**Nice work with potential, but contributions need to be strengthened**

**Confidence:** 5
**Recommendation:** Reject

**Metareview:**

The paper proposes to use importance resampling (IR) as an alternative to the more popular importance sampling (IS) approach to off-policy RL.  The hope is to reduce variance, as shown in experiments.  However, there is no analysis why/when IR will be better than IS for variance reduction, and a few baselines were suggested by reviewers.  While the authors rebuttal was helpful in clarifying several issues, the overall contribution does not seem strong enough for ICLR, on both theoretical and empirical sides.

The high variance of IS is known, and the following work may be referenced for better 1st order updates when IS weights are used: Karampatziakis & Langford (UAI'11).

In section 3, the paper says that most off-policy work uses d_mu, instead of d_pi, to weigh states.  This is true, but in the current context (infinite-horizon RL), there are more recent works that should probably be referenced:
  http://proceedings.mlr.press/v70/hallak17a.html
  https://papers.nips.cc/paper/7781-breaking-the-curse-of-horizon-infinite-horizon-off-policy-estimation